# Daptomycin avoids drug resistance mediated by the BceAB transporter in *Streptococcus pneumoniae*

Agathe Faure,[1] Sylvie Manuse,[1] Mathilde Gonin,[1] Christophe Grangeasse,[1] Jean-Michel Jault,[1] Cédric Orelle[1]

**ABSTRACT**    Drug-resistant bacteria are a serious threat to human health as antibiotics are gradually losing their clinical efficacy. Comprehending the mechanism of action of antimicrobials and their resistance mechanisms plays a key role in developing new agents to fight antimicrobial resistance. The lipopeptide daptomycin is an antibiotic that selectively disrupts Gram-positive bacterial membranes, thereby showing slower resistance development than many classical drugs. Consequently, it is often used as a last resort antibiotic to preserve its use as one of the least potent antibiotics at our disposal. The mode of action of daptomycin has been debated but was recently found to involve the formation of a tripartite complex between undecaprenyl precursors of cell wall biosynthesis and the anionic phospholipid phosphatidylglycerol. BceAB-type ABC transporters are known to confer resistance to antimicrobial peptides that sequester some precursors of the peptidoglycan, such as the undecaprenyl pyrophosphate or lipid II. The expression of these transporters is upregulated by dedicated two-component regulatory systems in the presence of antimicrobial peptides that are recognized by the system. Here, we investigated whether daptomycin evades resistance mediated by the BceAB transporter from the bacterial pathogen *Streptococcus pneumoniae*. Although daptomycin can bind to the transporter, our data showed that the BceAB transporter does not mediate resistance to the drug and its expression is not induced in its presence. These findings show that the pioneering membrane-active daptomycin has the potential to escape the resistance mechanism mediated by BceAB-type transporters and confirm that the development of this class of compounds has promising clinical applications.

**IMPORTANCE**    Antibiotic resistance is rising in all parts of the world. New resistance mechanisms are emerging and dangerously spreading, threatening our ability to treat common infectious diseases. Daptomycin is an antimicrobial peptide that is one of the last antibiotics approved for clinical use. Understanding the resistance mechanisms toward last-resort antibiotics such as daptomycin is critical for the success of future antimicrobial therapies. BceAB-type ABC transporters confer resistance to antimicrobial peptides that target precursors of cell-wall synthesis. In this study, we showed that the BceAB transporter from the human pathogen *Streptococcus pneumoniae* does not confer resistance to daptomycin, suggesting that this drug and other calcium-dependent lipopeptide antibiotics have the potential to evade the action of this type of ABC transporters in other bacterial pathogens.

**KEYWORDS**    antibiotic resistance, antimicrobial peptides, ABC transporters, two-component regulatory systems, antibiotics, *Streptococcus pneumoniae*

Several cyclic lipopeptide antibiotics have been discovered, but apart from polymyxins and antifungal echinocandins, daptomycin is the only one to have reached clinical approval (1). Daptomycin is a cyclic antimicrobial lipopeptide that was originally isolated in the 1980s from the Gram-positive soil actinomycete *Streptomyces roseosporus*.

Address correspondence to Jean-Michel Jault, jean-michel.jault@ibcp.fr, or Cédric Orelle, cedric.orelle@ibcp.fr.

The authors declare no conflict of interest.

See the funding table on p. 10.

It was the first in class of a novel group of calcium-dependent lipopeptides with potent activity against Gram-positive bacteria (1). The commercialization of daptomycin is one of the last antibiotic classes introduced into the market (2). First approved in 2003 for the treatment of complicated skin and skin-structure infections, daptomycin was subsequently approved in 2006 for the treatment of right-sided endocarditis and bacteremia (3). Remarkably, daptomycin is one of the few peptide antibiotics that can be administered systemically and is generally prescribed as a last-resort antimicrobial agent in treating severe infections due to Gram-positive pathogens. These include drug-resistant bacteria such as methicillin-resistant and vancomycin-resistant *Staphylococcus aureus* (MRSA & VRSA) and vancomycin-resistant *Enterococci*.

Membrane-active antibiotics thus hold great promise for slower resistance development and have recently attracted renewed interest in drug development. Daptomycin's mechanism of action has been highly debated and many controversial results accumulated over the years did not permit to clearly establish its molecular target. Recently, however, $Ca^{2+}$-daptomycin was shown to specifically interact with undecaprenyl precursors of cell walls in the presence of the anionic phospholipid phosphatidylglycerol (PG), forming a tripartite complex (4). These precursors of cell wall synthesis include undecaprenyl-pyrophosphate (C55-PP or UPP), undecaprenyl-phosphate (C55-P or UP), and lipid II, which are also targeted by a number of other antimicrobial peptides such as bacitracin for UPP (5–9). Daptomycin-resistant phenotypes can be linked to alterations in the composition of the bacterial membrane and changes in cell wall biosynthesis (10–12). Consistent with the daptomycin mode of action, a common resistance mechanism involves the alteration of the cell surface charge leading to the repulsion of daptomycin molecules, or changes related to the bacterial phospholipid phosphatidylglycerol (PG). For instance, daptomycin resistance in *S. aureus* is classically associated with mutations in *mprF*, which encodes a bifunctional membrane protein that performs lysylation of PG, effectively masking PG on the membrane (3). Another known resistance mechanism involves the alteration of complex transcriptional regulatory networks involved in the cell wall stress response and membrane homeostasis. For instance, transcriptional changes of the *walKR* or *vraSR* two-component regulatory systems (TCS) have been linked to increased daptomycin resistance in *S. aureus* (13). It was recently suggested that BceAB-type ABC transporters, whose expressions are also regulated by dedicated TCS systems (14, 15), provide resistance *via* target protection by transiently releasing lipid II cycle intermediates from the inhibitory sequestration of antimicrobial peptides (16). While many Gram-positive bacteria contain up to six BceAB-type transporters with both distinct and overlapping substrate specificities (17), *Streptococcus pneumoniae* contains a single one that provides resistance to a large number of antimicrobial peptides targeting lipid II (18). In this study, we investigated whether this BceAB transporter confers resistance to daptomycin in *S. pneumoniae*. This bacterium causes each year over one million deaths in the world, mostly in children, elderly, and immunocompromised people. It usually colonizes the human nasopharynx but can sometimes invade distant sites, causing infections such as sinusitis and otitis media but also life-threatening invasive diseases such as pneumonia, meningitis, and septicemia (19, 20). As for most pathogenic bacteria, antibiotic resistance is an increasing issue in pneumococcal infections (21). In 2017, the WHO listed *S. pneumoniae* as one of the 12 priority targets for research and development of new antibiotics. Currently, this pathogen is the fourth leading pathogen for deaths associated with resistance, behind *Escherichia coli*, *Staphylococcus aureus*, and *Klebsiella pneumoniae* (21). Although daptomycin is strongly active against *S. pneumoniae in vitro* (22–24), it is inhibited by lung surfactant (25) and hence is not effective for the treatment of pneumonia (26, 27). However, daptomycin is highly effective in preventing *S. pneumoniae*-induced septic death (26) and in a rabbit model of meningitis (28). Hence, understanding the mechanism and molecular resistance determinants toward last-resort antibiotics such as daptomycin is important to optimize future antimicrobial therapies against bacterial pathogens, including *Streptococcus pneumoniae*.

## RESULTS

### Analysis of daptomycin resistance mediated by the BceAB/TCS01 system in *S. pneumoniae*

In *S. pneumoniae*, the BceAB-type ABC transporter confers resistance to a number of antimicrobial peptides targeting the undecaprenyl-pyrophosphate or the lipid II (18). These molecules include bacitracin, nisin, actagardin, planosporicin, or NAI-802. We previously demonstrated that the expression of the *bceAB* genes is upregulated by TCS01 in the presence of the antimicrobial peptides that are recognized by the system (Fig. 1). The ΔbceAB and Δtcs01 strains displayed the same increased sensitivity to the aforementioned AMPs as compared to the wild-type strain. Here, using a similar strategy, we investigated whether BceAB and/or TCS01 provide resistance to daptomycin. We used the broth microdilution method to determine the Minimum Inhibitory Concentration (MIC) of daptomycin for the wild-type and the two mutant strains ΔbceAB and Δtcs01. The experiments were first conducted with the R6 avirulent strain that lacks a polysaccharide capsule and that is classically used for the investigation of pneumococcal biology (29). First, and as reported before (18), the deletion of *bceAB* and/or *tcs01* genes did not impair bacterial growth in Todd-Hewitt broth supplemented with 0.5% yeast extract (THY) (Fig. 2). In the presence of increasing concentrations of daptomycin, the growth of the mutant and wild-type strains remained similar, and the same MIC was determined at 4 µg/mL (Fig. 2 and Table 1). Similar results were observed with the R800 strain, which is a derivative of the R6 strain that contains a streptomycin-resistant mutation to facilitate the selection of the constructs of interest (30), although the MIC for daptomycin was 8 µg/mL (Fig. 3 and Table 1). These results indicate that the absence of BceAB nor TCS01 does not modulate pneumococcal sensitivity to daptomycin.

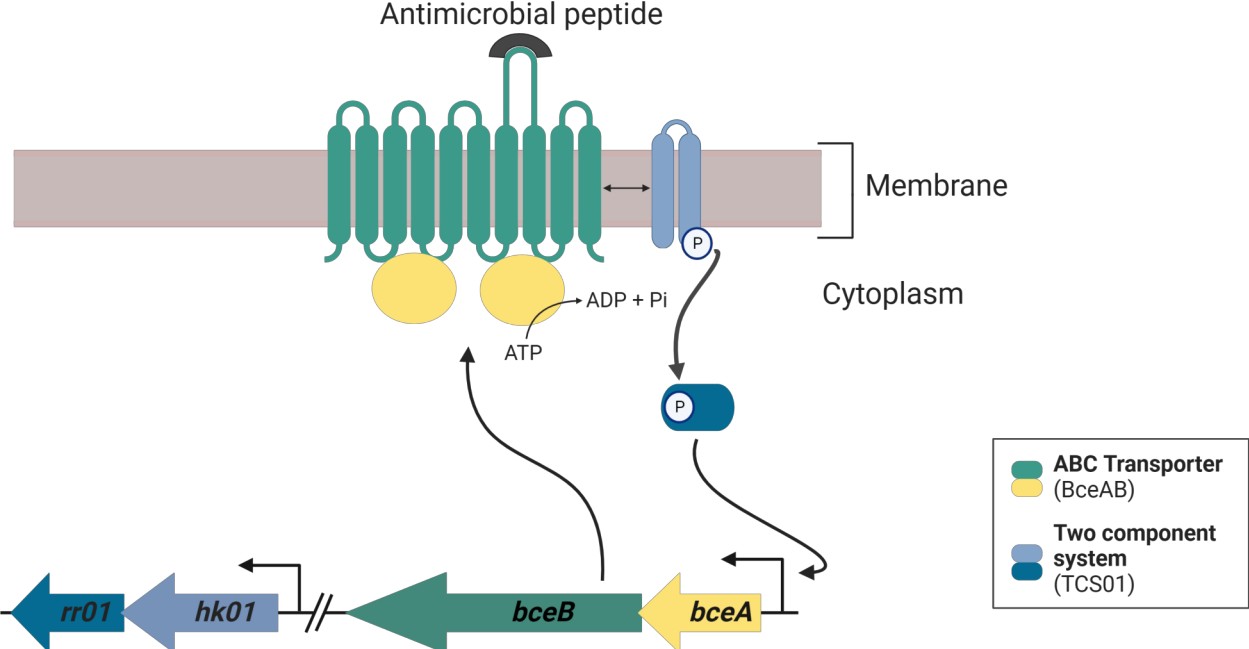

**FIG 1** Functional module involving TCS01 and BceAB in *Streptococcus pneumoniae*. In the absence of antimicrobial peptides, our current understanding is mostly based on work performed on the homologous system in *Bacillus subtilis*. BceAB and the histidine kinase likely form a relatively stable complex (31), and the transporter maintains its cognate kinase in an inactive state in the absence of bacitracin (32). Recognition of antimicrobial peptides and ATP binding/hydrolysis by BceAB are necessary to trigger phosphorelay signaling through BceS (33) (HK01 in *S. pneumoniae*). This signaling involves the autophosphorylation of the histidine kinase, and the phosphoryl group is then passed on to a conserved aspartate residue of the response regulator. Our previous work showed that the BceAB/TCS01 from the *S. pneumoniae* module can recognize the presence of antimicrobial peptides targeting undecaprenylpyrophosphate (UPP) or lipid II. In the presence of antimicrobial peptides, the phosphorylated RR01 upregulates the expression of the operon containing the *bceAB* genes but does not regulate the *tcs01* operon (18). The overexpression of BceAB mediates antimicrobial peptide resistance. Figure created with Biorender.com.

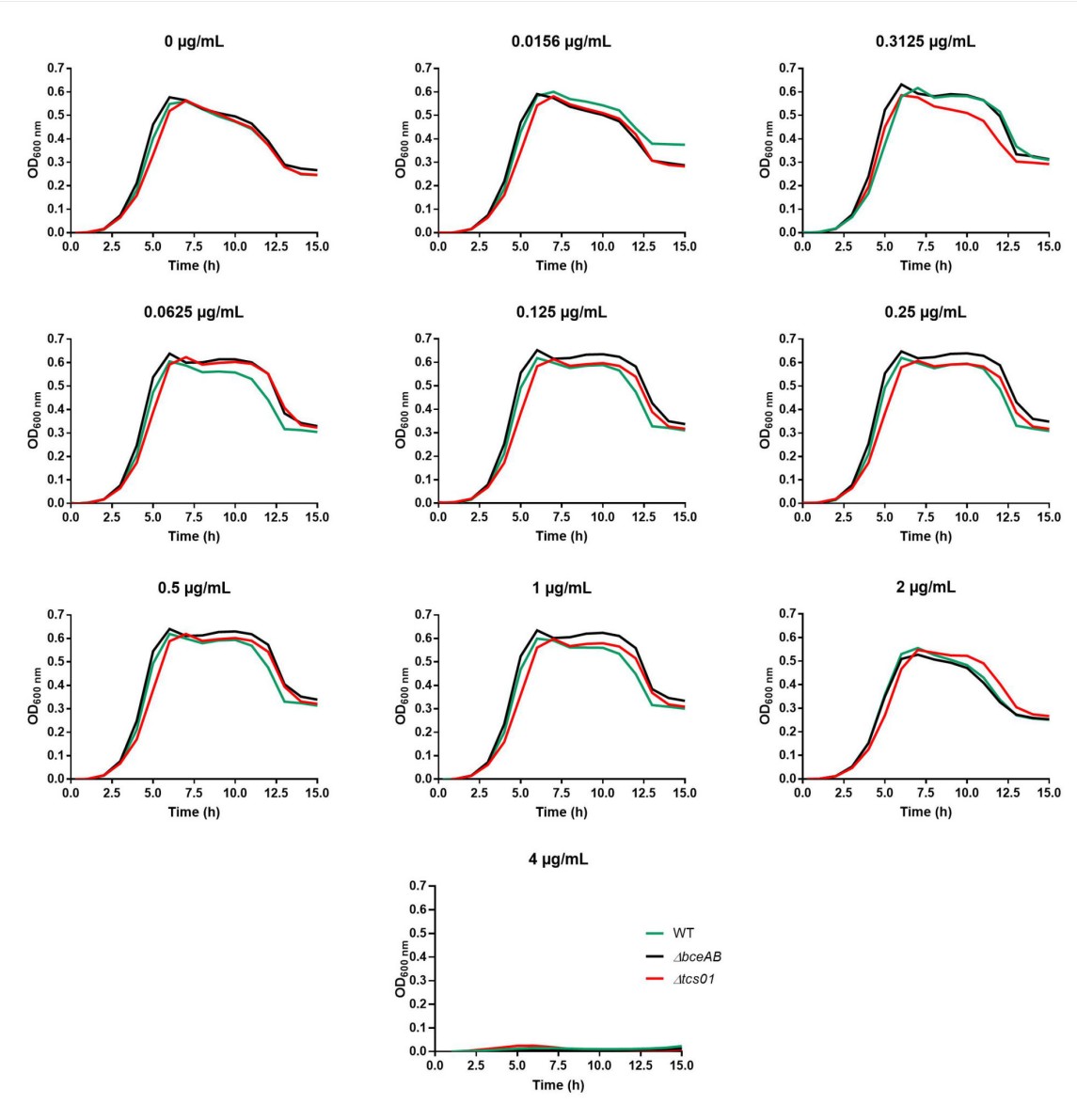

**FIG 2** Growth of wild-type and mutant R6 strains in the absence or presence of various concentrations of daptomycin. The daptomycin concentrations are indicated on top of the graphs. Cultures were performed in microplates, in the presence of 50 µg/mL of CaCl₂. Data shown here are the average of technical duplicates. These experiments were confirmed with biological replicates shown in Table 1.

## Daptomycin does not significantly induce the expression of the *bceAB* genes

To analyze the expression of the BceAB transporter when *S. pneumoniae* is challenged by antimicrobial peptides, we previously engineered a R800 strain in which the *gfp* gene was fused to the 3′-end of the gene encoding the nucleotide-binding domain BceA at its endogenous chromosomal location. Importantly, the GFP fusion did not impair the functionality of BceAB since the engineered strain displayed comparable resistance for bacitracin, nisin, and actagardin as compared to the wild-type strain (18). We submitted *S. pneumoniae* cells to either bacitracin/actagardin as controls or daptomycin and analyzed the levels of GFP fluorescence in each sample after migration on a SDS-PAGE since GFP is not denatured in SDS-polyacrylamide gels run under certain conditions (34, 35). In contrast to bacitracin or actagardin treatment, subinhibitory concentrations of daptomycin failed to increase the expression of *bceAB-gfp* (Fig. 4). To

**TABLE 1** Minimum inhibitory concentrations (µg/mL) of daptomycin against the R6 and R800 strains[a]

| | MIC (µg/mL) | | | Fold change | |
|---|---|---|---|---|---|
| **Strain R6** | | | | | |
| WT | Δ*bceAB* | Δ*tcs01* | Δ*bceAB* | | Δ*tcs01* |
| 4 | 4 | 4 | 1 | | 1 |
| **Strain R800** | | | | | |
| WT | Δ*bceAB* | Δ*hk01* | Δ*bceAB* | | Δ*hk01* |
| 8 | 8 | 8 | 1 | | 1 |

[a]In the R6 strain, the whole operon of the *tcs01* was deleted (Δ*tcs01*) while in the R800 strain, only the histidine kinase gene was deleted (Δ*hk01*). Data correspond to four biological replicates, each of them with technical duplicates.

complement this global analysis, we further quantified the production of BceAB-GFP at the single-cell level by fluorescence microscopy. While bacitracin substantially increased the expression of the BceAB transporter, as shown before (18), actagardin induced the overexpression of the transporter at much higher levels (Fig. 5). By contrast, various subinhibitory concentrations of daptomycin failed to upregulate the expression of the BceAB transporter.

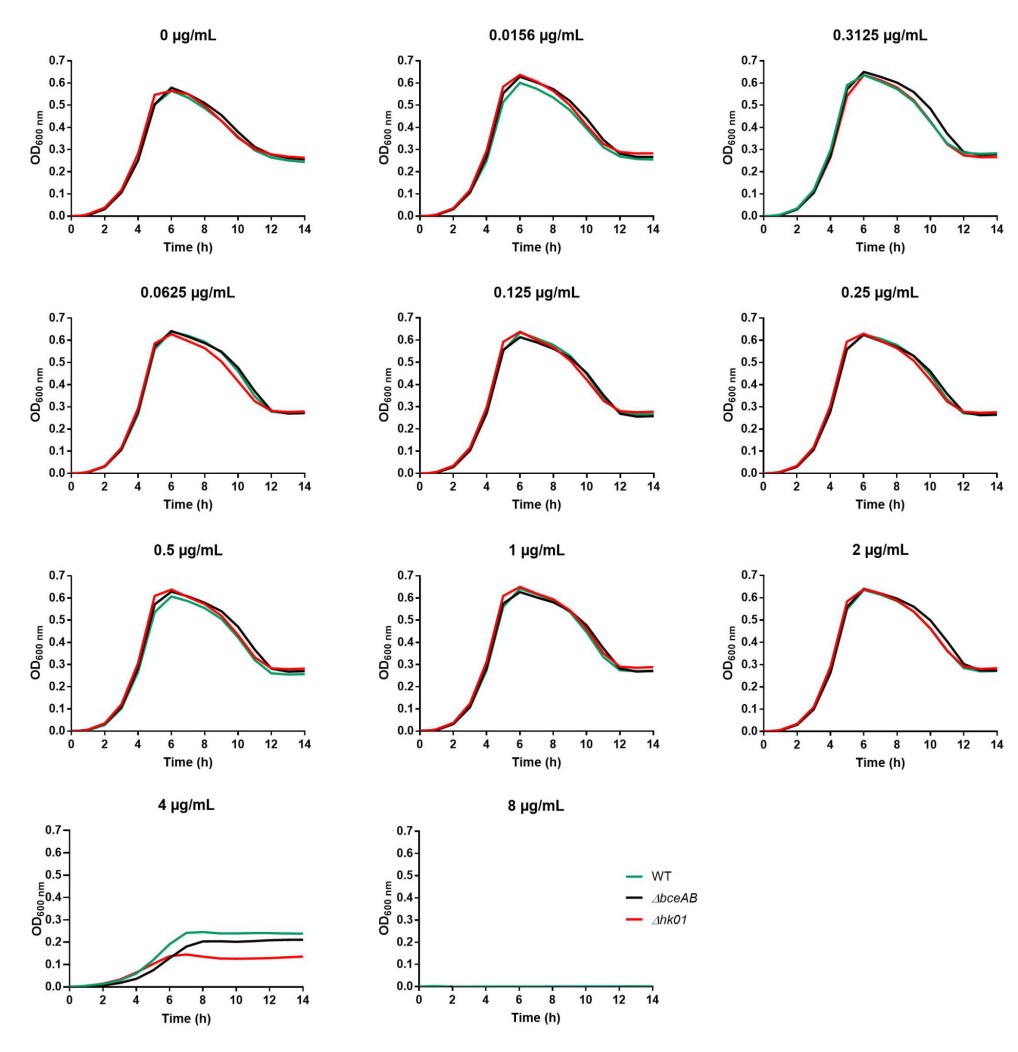

**FIG 3** Growth of wild-type and mutant R800 strains in the absence or presence of various concentrations of daptomycin. The daptomycin concentrations are indicated on top of the graphs. Cultures were performed in microplates, in the presence of 50 µg/mL of $CaCl_2$. Data shown here are the average of technical duplicates. These experiments were confirmed with biological replicates shown in Table 1.

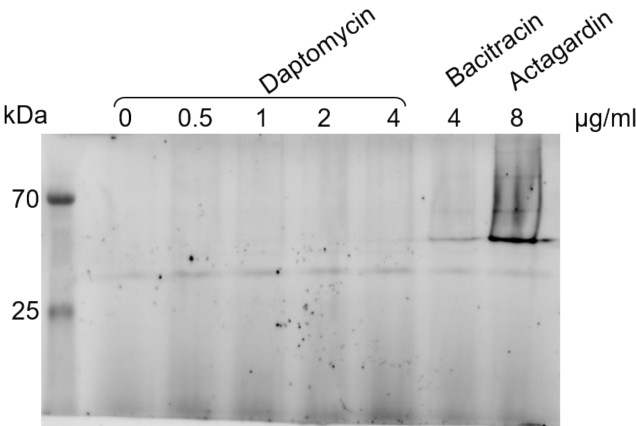

**FIG 4** Visualization of BceA-GFP fluorescence on the gel. The culture of the R800 *bceA-gfp* strain was performed until OD$_{600nm}$ = 0.3. Bacteria were incubated for 30 min in the presence of different concentrations of daptomycin (in the presence of CaCl$_2$). Incubation in the presence of bacitracin (4 µg/mL) or the presence of actagardin (8 µg/mL) was used as positive control. Once treated, the bacterial pellets were loaded onto a 15% SDS-PAGE gel. In-gel fluorescence of BceA-GFP was scanned with a typhoon imager.

## Interaction of daptomycin with the BceAB transporter

We previously showed that we could express this transporter in *E. coli* (36) and purify it to homogeneity (18). After reconstitution in proteoliposomes, the transporter displays a substantial ATPase activity that can be stimulated up to twofold by antimicrobial

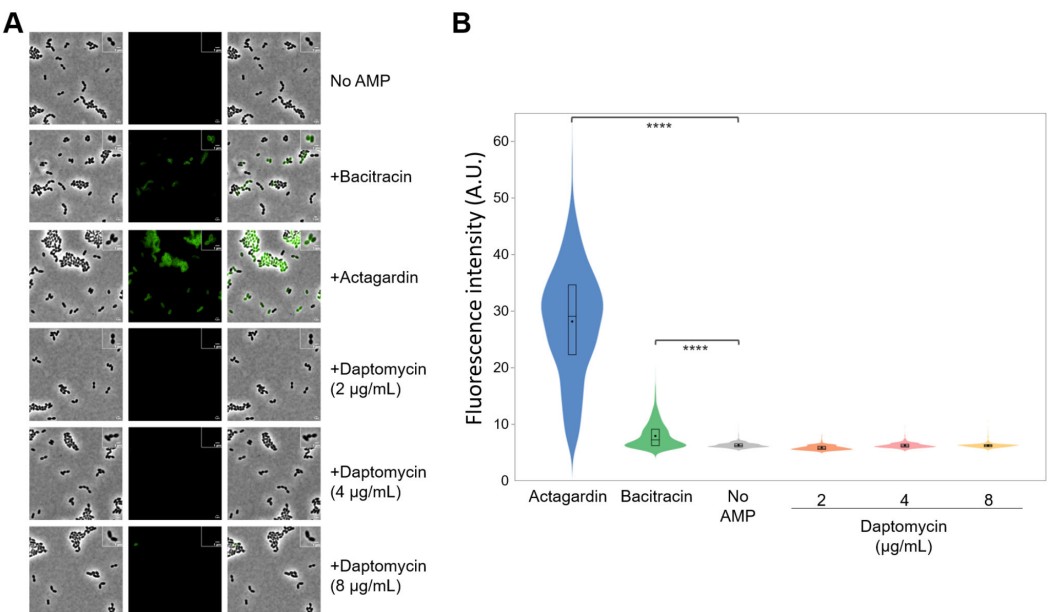

**FIG 5** Analysis of BceAB-GFP expression in *S. pneumoniae* upon daptomycin treatment. These experiments were conducted in the R800 *bceAB-gfp* strains. Bacitracin (1 µg/mL) and actagardin (8 µg/mL) are used here as positive controls. Three daptomycin concentrations were tested: 2 µg/mL, 4 µg/mL, and 8 µg/mL. (A) Fluorescence microscopy of *S. pneumoniae* cells analyzing the drug-dependent expression of BceAB-GFP. Phase contrast (left panel), GFP fluorescent signal (middle panel), and overlays between phase contrast and GFP images (right panel) are shown. Enlargement is shown on the upper right corners of each panel. Scale bar, 1 µm. (B) Violin plots showing the distribution of cellular fluorescence mean intensities in individual cells corresponding to panel A. The boxes in the violin plots indicate the 25th to the 75th percentiles and the whiskers indicate the minimum and maximum value. The mean and the median are indicated with a dot and a line in the box, respectively. Significance was determined using Kruskal–Wallis and Dunn's multiple comparison tests. A total of 11,185 cells were analyzed: actagardin $n$ = 2,153; bacitracin $n$ = 2,024; no drug $n$ = 2,023; daptomycin 2 µg/mL $n$ = 2,013, 4 µg/mL $n$ = 1,673, 8 µg/mL $n$ = 1,299.

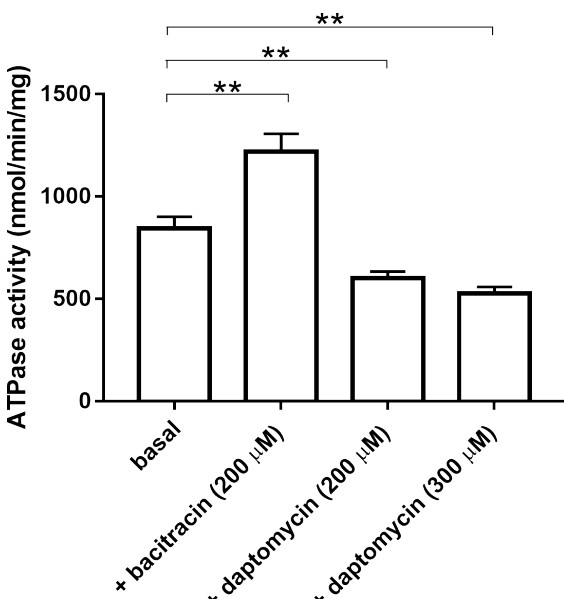

**FIG 6** ATPase activities of the BceAB transporter reconstituted in proteoliposomes. Data are the average of three replicates and error bars indicate the standard deviation of the replicates. Statistical significance was calculated by Student´s $t$-test with Welch's correction between the conditions indicated with brackets. Statistically significant differences are indicated with ** ($P \leq 0.01$).

peptides to which it confers resistance to, while the other AMPs either do not affect or inhibit the ATPase activity (18). Since ATP and AMPs interact on opposite sides of the transporter, we used experimental conditions that permeabilize the lipid vesicles (10 mM $Mg^{2+}$, 37°C). Such a strategy was successfully employed to measure the substrate-stimulated ATPase activities of various ABC importers in proteoliposomes since their activities strongly depend on a substrate-binding protein interacting on the extracellular side of the membrane (37, 38). Here, we sought to determine whether the ATPase activity of the transporter is sensitive to daptomycin, which would indicate that daptomycin can bind to the transporter without inducing resistance. We observed that daptomycin could inhibit 30%–40% of the basal ATPase activity (Fig. 6), suggesting that the drug can indeed bind to the transporter without eliciting induction and resistance, as previously shown for ramoplanin in our previous study (18).

## DISCUSSION

Whereas many Gram-positive bacteria contain up to six of these BceAB-type transporters with both distinct and overlapping substrate specificities (17), *S. pneumoniae* contains a single one that displays a relatively wide specificity. For instance, YvcRS (renamed PsdAB) confers resistance to nisin while BceAB allows resistance to actagardin and bacitracin in *Bacillus subtilis*. By contrast, BceAB in *S. pneumoniae* promotes resistance to the three AMPs. While BceAB-type transporters can recognize structurally unrelated AMPs, they can also display a clear selectivity between similar compounds; hence, the selectivity of individual BceAB-type transporters is hardly predictable. Nevertheless, the molecules that are recognized by these transporters mostly target precursors of cell wall biosynthesis, that is, UPP or lipid II. Because daptomycin likely targets undecaprenyl precursors in the presence of the anionic phospholipid PG, we investigated here whether this antibiotic is subject to AMP resistance mediated by the BceAB transporter in *S. pneumoniae*. Little is known about daptomycin resistance mediated by BceAB-type transporters and, to the best of our knowledge, such resistance was only reported with the transporter VraDEH from *S. aureus* (39). VraH is an additional small membrane protein that likely interacts with the BceAB transporter VraDE to modulate its substrate

specificity, and only VraDEH provides high levels of daptomycin resistance. No homolog of VraH is present in *Streptococus pneumoniae*, which is consistent with our results. Indeed, we showed that the BceAB transporter of *S. pneumoniae* does not confer resistance to daptomycin and that this antibiotic is not able to induce the expression of the transporter. Possibly the binding of daptomycin to the large extracellular domain of BceAB fails to stimulate the ATPase activity of the transporter since a functional ATPase activity is essential for the upregulation of its genes and the establishment of resistance (33, 40). Although sensing/signaling and resistance are functionally interconnected, there is not always a good correlation between the strength of induction and the rate of resistance conferred by the induced BceAB transporters. For instance, actagardin is the strongest inducer of the PsdRS-AB system in *B. subtilis*, but this system does not confer any detectable resistance (41). This observation suggests that sensing and target removal of AMPs are two separable functions of BceAB-like transporters (42). This assumption has been verified for BceAB in *B. subtilis* since several mutations could be identified that strongly affected one activity of the transporter but with only minor effects on the other (43). The fact that daptomycin does not trigger the expression of the BceAB transporter from *S. pneumoniae* was important to address because the use of daptomycin would not induce resistance toward other antimicrobial peptides or antibiotics. Combinations of antibiotics is an effective strategy to fight drug-resistant bacteria (44) but their optimization requires a deep understanding of the molecular mechanisms underpinning their action since their combinations and dosages have been largely determined empirically in clinical settings (45). It is also noteworthy that the signaling pathway initiated by BceAB involves the activation of TCS01 (Fig. 1), which strongly contributes to pneumococcal virulence in several infection models (46–48). Pioneering work has notably evidenced that disruption of *tcs01* genes causes a dramatic attenuation of the growth (by $10^5$ fold) in a mouse respiratory tract infection model (47). Therefore, daptomycin might be a last-resort drug of choice in the treatment of severe complications of pneumococcus diseases, including bacteremia and septicemia (26). The rise of antibiotic resistance is a global concern that threatens to undermine many aspects of modern medicine. Addressing this menace will require the discovery and development of new antibiotics that operate by unexploited modes of action and escape common resistance mechanisms. The calcium-dependent lipopeptide antibiotics are an important emerging class of natural products that provide a source of mechanistically diverse antimicrobial agents (49). The fact that daptomycin is not subject to the antimicrobial peptide resistance mediated by the BceAB transporter from *S. pneumoniae* is promising since this drug may also escape the action of this type of ABC transporter in other bacterial pathogens. Investigation of these aspects and other daptomycin resistance mechanisms will certainly increase our global knowledge and contribute to the optimization of future treatments.

## MATERIALS AND METHODS

### Source of antimicrobial peptides

The AMPs were purchased from Sigma-Aldrich (Bacitracin), Adipogen (Actagardin), and TOKU-E (Daptomycin).

### Bacterial strains and growth

Bacterial strains are from reference (18). Cells were grown at 37°C in a Todd-Hewitt medium supplemented with yeast extract (0.5%—THY), in an anaerobic condition, without agitation.

## MIC determination

The MIC of various antimicrobial peptides for R6 and R800 strains was determined by the classical twofold broth dilution method. Cells were first grown in THY medium at 37°C without agitation and in anaerobic conditions until the absorbance reached approximately 0.3. Cells were then diluted to a final $OD_{600\ nm}$ = 0.002 in 96-well plates containing 300 µL of THY medium with serial dilutions of antimicrobial peptides. Pneumococcal growth was monitored by a microplate reader (TECAN) at 37°C without agitation. The absorbance was followed at 600 nm every 15 min for 15 h.

## Microscopy

Pneumococcal cells were grown at 37°C until $OD_{600nm}$ = 0.3 in C + Y medium (50). Cells were incubated with or without antimicrobial peptide for 30 min at 37°C, without agitation. Cells were imaged with a Nikon TiE fitted with an Orca-CMOS Flash4 V2 camera with a 100×/1.45 numerical aperture objective Images were collected using NIS-Elements (Nikon) and analyzed using ImageJ (http://rsb.info.nih.gov/ij/) and the plugin MicrobeJ (51). For each cell in each panel, the mean fluorescence intensity (a.u.) was automatically extracted and plotted with MicrobeJ to generate violin plots. Statistical analysis was performed using Prism 9 (GraphPad software).

## Typhoon imaging

The strains were cultured in 15 mL THY until $OD_{600nm}$ = 0.3. One mL of culture was incubated without or with antimicrobial peptides for 30 min at 37°C. The cultures ($OD_{600nm}$ ~ 0.6 for the untreated and daptomycin samples, and $OD_{600nm}$ ~ 0.44 for the actagardin and bacitracin and positive controls) were centrifuged for 15 min at $7,000 \times g$ and the pellet was recovered in 20 µL of 1× Laemmli without SDS or β-mercaptoethanol. The samples were plated on an 18% SDS-PAGE and migrated in a cold chamber at 180 V. The fluorescence of the gel was revealed using fluorescence imaging (TyphoonTM FLA 9500).

## Purification and reconstitution of the BceAB transporter

The transporter was expressed and purified as previously described (18). The proteoliposomes were essentially prepared as described for the ABC transporter BmrA (52). Eighty microliters of *Escherichia coli* phospholipids total extract (Avanti Polar lipids) at 25 mg/mL (water) was incubated under stirring at room temperature with 20 µL of 10% *n*-dodecyl-β-D-maltoside (Anatrace). After 1 h, 100 µg of protein was added in a final volume of 500 µL (50 mM HEPES/KOH, pH 8). After 45 min incubation, three successive additions of 40 mg Bio-Beads SM2 (Bio-Rad) were performed every hour. The proteoliposome solution was removed from the Biobeads and kept at 4°C.

## ATPase assays

Since ATP and antimicrobial peptides interact on opposite sides of the transporter, we used experimental conditions that permeabilize the proteoliposomes (10 mM $Mg^{2+}$, 37°C), as originally described (37). Activities were measured in 700 µL total volume in buffer HEPES-KOH 50 mM pH 8, 10 mM $MgCl_2$, 4 mM phosphoenolpyruvate, 32 µg/mL of lactate dehydrogenase, 60 µg/mL pyruvate kinase, and 5 mM ATP. The buffer was heated at 37°C for 5 min before adding the protein. Activities of 2 µg BceAB in proteoliposomes were then measured by absorbance at 340 nm for 20 min at 37°C.

## ACKNOWLEDGMENTS

This work was supported by the "Agence Nationale de la Recherche" (ANR-17-CE11-0045-01 to CO) and the Auvergne-Rhône-Alpes R&D Booster (Call 2021, project IMABGEN).

## AUTHOR AFFILIATION

[1]Molecular Microbiology and Structural Biochemistry (MMSB), UMR 5086 CNRS/University of Lyon, Lyon, France

## AUTHOR ORCIDs

Jean-Michel Jault  http://orcid.org/0000-0003-1743-2777
Cédric Orelle  http://orcid.org/0000-0003-3418-3290

## FUNDING

| Funder | Grant(s) | Author(s) |
| --- | --- | --- |
| Agence Nationale de la Recherche (ANR) | ANR-17-CE11-0045-01 | Cédric Orelle |
| Region Rhone-Alpes | R&D Booster 2021 | Jean-Michel Jault |
| | | Cédric Orelle |

## AUTHOR CONTRIBUTIONS

Agathe Faure, Data curation, Formal analysis, Investigation, Writing – review and editing | Sylvie Manuse, Data curation, Formal analysis, Investigation, Supervision, Validation, Writing – review and editing | Mathilde Gonin, Formal analysis, Investigation | Christophe Grangeasse, Supervision, Validation, Writing – review and editing | Jean-Michel Jault, Conceptualization, Supervision, Validation, Writing – review and editing | Cédric Orelle, Conceptualization, Formal analysis, Funding acquisition, Project administration, Supervision, Validation, Writing – original draft

## ADDITIONAL FILES

The following material is available online.

Open Peer Review

**PEER REVIEW HISTORY (review-history.pdf).** An accounting of the reviewer comments and feedback.

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
