## [Reviewer comments · Microbiology Spectrum]

Microbiology Spectrum

Daptomycin avoids drug resistance mediated by the BceAB transporter in *Streptococcus pneumoniae*

Agathe Faure, Sylvie Manuse, Mathilde Gonin, Christophe Grangeasse, Jean-Michel Jault, and Cédric Orelle

Corresponding Author(s): Cédric Orelle, Centre National de la Recherche Scientifique

Review Timeline:

Submission Date:	October 12, 2023
Editorial Decision:	November 29, 2023
Revision Received:	December 15, 2023
Accepted:	December 19, 2023

Editor: Eric Cascales

Reviewer(s): The reviewers have opted to remain anonymous.

Transaction Report:

DOI: <https://doi.org/10.1128/spectrum.03638-23>

Re: Spectrum03638-23 (Daptomycin avoids drug resistance mediated by the BceAB transporter in *Streptococcus pneumoniae*)

Dear Cédric,

Thank you for submitting your work at Microbiology Spectrum. Your manuscript has been reviewed by two experts in the field. As you will see in their comments pasted below, the two reviewers had a very positive appraisal of your work, as I do. They have however raised a few concerns that should be easy to address, and notably the point #1 of reviewer #1 who is concerned by peptide binding onto BceAB reconstituted in liposomes. Based on the reviewer's opinion and recommendation, I encourage you to carefully address the questions raised by the reviewers, and invite you to submit a revised version of your work within 60 days; if you cannot complete the modification within this time period, please contact me. If you do not wish to modify the manuscript and prefer to submit it to another journal, notify me immediately so that the manuscript may be formally withdrawn from consideration by Spectrum.

Revision Guidelines

Sincerely,
Eric

Eric Cascales
Editor
Microbiology Spectrum

Reviewer #1 (Comments for the Author):

In this manuscript by Faure et al. the authors investigate whether or not the BceAB transporter from *Streptococcus pneumoniae*

senses and provides resistance to the clinically important antimicrobial peptide daptomycin. Bce type transporters and their associated two-component systems have been recognized for their ability to sense and provide resistance against various antimicrobial peptides across Gram-positive organisms. Despite a recent wealth of bioinformatic, structural, and functional research on these systems, much of the details governing substrate specificity and overlap between systems remains enigmatic. The authors have done an excellent job elaborating on these issues through the introduction and discussion. Further evaluation of the antimicrobial peptide specificity and overlap between Bce-type systems in various organisms is certainly needed, and the authors further expand our knowledge in this area by providing new insight into how the BceAB system in *Streptococcus pneumoniae* responds to daptomycin and other relevant peptides. Overall, I feel that the manuscript is well written, timely, and provides an important advance in knowledge for the field. I only have two major comments, and a few minor comments.

Major comments:

1. The authors rely on ATPase activity measurements to gauge whether or not the antimicrobial peptides can bind to the purified and liposome reconstituted BceAB transporter. During liposome reconstitution the BceAB transporter could theoretically be incorporated into the lipid bilayer in one of two orientations (ie: NBD in the lumen, or NBD facing out of the liposome). Only the transporter that is incorporated in the orientation with NBDs facing outside of the liposome should result in detectable ATPase activity, as the ATP-Mg, NADH, and other components of the ATPase reaction are not membrane penetrant. However, in this transporter orientation the putative antimicrobial peptide binding region would be oriented toward the liposome lumen. As the peptides bacitracin and daptomycin do not cross lipid membranes, it is difficult for me to envision how the added peptides could possibly reach their proposed binding site in the extracellular loop of the BceAB transporter (which is pointed toward the sealed interior of the liposome). Thus, I am slightly skeptical about whether or not the data in figure 6 truly demonstrates "binding" of the peptides to the transporter in a relevant fashion. Can the authors provide any explanation as to how bacitracin or daptomycin could reach their proposed binding site in the interior of the sealed liposome? Or is there any other method they could use to conclusively demonstrate that the peptides "bind" to the purified transporter?

2. The fluorescence microscopy in figure 5 nicely demonstrates that Actagradin strongly induces expression of the BceAB transporter, even much more strongly than bacitracin. It seems like Actagradin should also be included as a control in the in-gel GFP expression analysis in figure 4. Additionally, some type of loading control should be included with figure 4 to demonstrate that all lanes have the same amount of total protein loaded. This could be accomplished through a western blot of a known standard protein, or simply coomassie blue staining of the gel following fluorescence imaging.

Minor comments:

1. The cartoon in figure 1 gives a false sense of the relative sizes of different components of the system. For instance, in the current cartoon the RR is drawn very large relative to the transporter and HK. Similarly, the antimicrobial peptide is shown to be almost as large as the RR, when in reality it is significantly smaller. While I realize this is simply a cartoon illustration, it may be helpful for a more general audience to consider trying to draw the components to a more relative scale.

2. line 78 - delete the "s" from interacts

3. line 95 - "release" should be "releasing"

Reviewer #2 (Comments for the Author):

In attached document

To explore the potential resistance of the BceAB transporter to peptides that target precursors of cell wall synthesis precursors, this manuscript investigates the impact of daptomycin. Daptomycin- a cyclic antimicrobial lipopeptide effective against Gram positive bacteria, interacts with the undecaprenyl precursors of cell wall in the presence of the PG. Given that BceAB transporter confer resistance to antimicrobial peptides targeting lipids, the authors aimed to determine whether the BceAB transporter/TCS01 system also provides resistance to daptomycin in *S. pneumoniae* (which has a single copy of the BceAB transporter). As a defense mechanism BceAB is suspected to bind UPP and/or UPP antimicrobial complex, thus preventing the targeting of precursors. The data in the manuscript indicates that despite BceAB/TCS01 knockouts not rendering the bacteria more sensitive to daptomycin and the BceAB transporter not being upregulated in the presence of daptomycin, daptomycin does inhibit atpase activity.

1. Figure 1 needs a more detailed explanation in the text or figure legend- does HK form a complex with BceAB once phosphorylated- as represented by the arrow? Does it in turn phosphorylate RR, losing its phosphorylation? Come clarity on what is known, and speculation would be helpful in the model.

2. Curious on the impact of 4ug/mL daptomycin with delta bcsAB and delta hk01 since delta hk01 looks like it impacts growth more than delta bcsAB or WT. Do we see a more subtle difference at 3ug/mL (statistically significant difference)?

3. Unclear on the hypothesis for BcsAB activity, does daptomycin partially inhibit transport?

4. Please provide more details on purification, reconstitution in proteoliposomes and ATPase assays since references lead to another reference.

Review: 318694

Faure et al. Daptomycin avoids drug resistance mediated by the BceAB 1 transporter in *Streptococcus pneumoniae*

To explore the potential resistance of the BceAB transporter to peptides that target precursors of cell wall synthesis precursors, this manuscript investigates the impact of daptomycin. Daptomycin- a cyclic antimicrobial lipopeptide effective against Gram positive bacteria, interacts with the undecaprenyl precursors of cell wall in the presence of the PG. Given that BceAB transporter confer resistance to antimicrobial peptides targeting lipids, the authors aimed to determine whether the BceAB transporter/TCS01 system also provides resistance to daptomycin in *S. pneumoniae* (which has a single copy of the BceAB transporter). As a defense mechanism BceAB is suspected to bind UPP and/or UPP antimicrobial complex, thus preventing the targeting of precursors. The data in the manuscript indicates that despite BceAB/TCS01 knockouts not rendering the bacteria more sensitive to daptomycin and the BceAB transporter not being upregulated in the presence of daptomycin, daptomycin does inhibit atpase activity.

1. Figure 1 needs a more detailed explanation in the text or figure legend- does HK form a complex with BceAB once phosphorylated- as represented by the arrow? Does it in turn phosphorylate RR, losing its phosphorylation? Come clarity on what is known, and speculation would be helpful in the model.
2. Curious on the impact of 4ug/mL daptomycin with delta bcsAB and delta hk01 since delta hk01 looks like it impacts growth more than delta bcsAB or WT. Do we see a more subtle difference at 3ug/mL (statistically significant difference)?
3. Unclear on the hypothesis for BcsAB activity, does daptomycin partially inhibit transport?
4. Please provide more details on purification, reconstitution in proteoliposomes and ATPase assays since references lead to another reference.

Dear Cédric,

Thank you for submitting your work at Microbiology Spectrum. Your manuscript has been reviewed by two experts in the field. As you will see in their comments pasted below, the two reviewers had a very positive appraisal of your work, as I do. They have however raised a few concerns that should be easy to address, and notably the point #1 of reviewer #1 who is concerned by peptide binding onto BceAB reconstituted in liposomes. Based on the reviewer's opinion and recommendation, I encourage you to carefully address the questions raised by the reviewers, and invite you to submit a revised version of your work within 60 days; if you cannot complete the modification within this time period, please contact me. If you do not wish to modify the manuscript and prefer to submit it to another journal, notify me immediately so that the manuscript may be formally withdrawn from consideration by Spectrum.

Sincerely,
Eric

Eric Cascales
Editor
Microbiology Spectrum

Dear Editor,

Thank you for handling our manuscript. We really appreciated the constructive comments of the reviewers. Our point-by-point answers to the referees are provided in blue, and we hope that our revised manuscript will be judged suitable for publication in Microbiology Spectrum.

Reviewer #1 (Comments for the Author):

In this manuscript by Faure et al. the authors investigate whether or not the BceAB transporter from *Streptococcus pneumoniae* senses and provides resistance to the clinically important antimicrobial peptide daptomycin. Bce type transporters and their associated two-component systems have been recognized for their ability to sense and provide resistance against various antimicrobial peptides across Gram-positive organisms. Despite a recent wealth of bioinformatic, structural, and functional research on these systems, much of the details governing substrate specificity and overlap between systems remains enigmatic. The authors have done an excellent job elaborating on these issues through the introduction and discussion. Further evaluation of the antimicrobial peptide specificity and overlap between Bce-type systems in various organisms is certainly needed, and the authors further expand our knowledge in this area by providing new insight into how the BceAB system in *Streptococcus pneumoniae* responds to daptomycin and other relevant peptides. Overall, I feel that the manuscript is well written, timely, and provides an important advance in knowledge for the field. I only have two major comments, and a few minor comments.

We are very grateful for your positive evaluation.

Major comments:

1. The authors rely on ATPase activity measurements to gauge whether or not the antimicrobial peptides can bind to the purified and liposome reconstituted BceAB transporter. During liposome reconstitution the BceAB transporter could theoretically be incorporated into the lipid bilayer in one of two orientations (ie: NBD in the lumen, or NBD facing out of the liposome). Only the transporter that is incorporated in the orientation with NBDs facing outside of the liposome should result in detectable ATPase activity, as the ATP-Mg, NADH, and other components of the ATPase reaction are not membrane penetrant. However, in this transporter orientation the putative antimicrobial peptide binding region would be oriented toward the liposome lumen. As the peptides bacitracin and daptomycin do not cross lipid membranes, it is difficult for me to envision how the added peptides could possibly reach their proposed binding site in the extracellular loop of the BceAB transporter (which is pointed toward the sealed interior of the liposome). Thus, I am slightly skeptical about whether or not the data in figure 6 truly demonstrates "binding" of the peptides to the transporter in a relevant fashion. Can the authors provide any explanation as to how bacitracin or daptomycin could reach their proposed binding site in the interior of the sealed liposome? Or is there any other method they could use to conclusively demonstrate that the peptides "bind" to the purified transporter?

We understand the concern of the reviewer and this is a valid question. In the experimental conditions that we are using, the proteoliposomes are not tightly sealed. This strategy was first used by the laboratory of Ames with the histidine ABC permease (Liu et al, J. Biol. Chem 1997 ; <https://doi.org/10.1074/jbc.272.35.21883>), and later for instance with the maltose ABC transporter (Orelle et al, PNAS 2008). Such strategy was necessary to reveal the substrate-stimulated ATPase of these importers, because the ATP and the substrate-binding protein, which is required to stimulate the ATPase activity, interact on opposite sides of the transporters. The trick is to use 10 mM of magnesium at 37 °C in the assay in order to permeabilize the proteoliposomes. We added this explanation in the revised manuscript to make this point clearer.

2. The fluorescence microscopy in figure 5 nicely demonstrates that Actagradin strongly induces expression of the BceAB transporter, even much more strongly than bacitracin. It seems like Actagradin should also be included as a control in the in-gel GFP expression analysis in figure 4. Additionally, some type of loading control should be included with figure 4 to demonstrate that all lanes have the same amount of total protein loaded. This could be accomplished through a western blot of a known standard protein, or simply coomassie blue staining of the gel following fluorescence imaging.

According to the recommendation of the reviewer, Actagradin was included as control in the in-gel GFP expression analysis in Figure 4.

We are providing hereafter for the reviewer the coomassie blue staining of the gel following fluorescence imaging. We did not include it to the manuscript, because the loading buffer did not contain SDS to preserve the folding and fluorescence of the GFP, and the gel is thus not sufficiently presentable. However, we measured the OD of all the samples before pelleting the cells and loading them in the gel, and this information is now included in the Material and Methods section.

Minor comments:

1. The cartoon in figure 1 gives a false sense of the relative sizes of different components of the system. For instance, in the current cartoon the RR is drawn very large relative to the transporter and HK. Similarly, the antimicrobial peptide is shown to be almost as large as the RR, when in reality it is significantly smaller. While I realize this is simply a cartoon illustration, it may be helpful for a more general audience to consider trying to draw the components to a more relative scale.

Thank you for the suggestion, we modified Figure 1 according to your comment.

2. line 78 - delete the "s" from interacts

Correction made, thank you.

3. line 95 - "release" should be "releasing"

Correction made, thanks a lot.

Reviewer #2 (Comments for the Author):

In attached document

To explore the potential resistance of the BceAB transporter to peptides that target precursors of cell wall synthesis precursors, this manuscript investigates the impact of daptomycin. Daptomycin- a cyclic antimicrobial lipopeptide effective against Gram positive bacteria, interacts with the undecaprenyl precursors of cell wall in the presence of the PG. Given that BceAB transporter confer resistance to antimicrobial peptides targeting lipids, the authors aimed to determine whether the BceAB transporter/TCS01 system also provides resistance to daptomycin in *S. pneumoniae* (which has a single copy of the BceAB transporter). As a defense mechanism BceAB is suspected to bind UPP and/or UPP antimicrobial complex, thus preventing the targeting of precursors. The data in the manuscript indicates that despite BceAB/TCS01 knockouts not rendering the bacteria more sensitive to daptomycin and the BceAB transporter not being upregulated in the presence of daptomycin, daptomycin does inhibit atpase activity.

1. Figure 1 needs a more detailed explanation in the text or figure legend- does HK form a complex with BceAB once phosphorylated- as represented by the arrow? Does it in turn phosphorylate RR, losing its phosphorylation? Come clarity on what is known, and speculation would be helpful in the model.

Thank you for your helpful suggestion. Figure 1 was modified accordingly

2. Curious on the impact of 4ug/mL daptomycin with delta bcsAB and delta hk01 since delta hk01 looks like it impacts growth more than delta bcsAB or WT. Do we see a more subtle difference at 3ug/mL (statistically significant difference)?

We also noticed that the growth of the delta hk01 is slightly impaired at 4ug/mL but we do not consider this effect significant since the MIC difference is <2.

3. Unclear on the hypothesis for BcsAB activity, does daptomycin partially inhibit transport?

Did you ask whether daptomycin can inhibit the resistance toward other AMPs ? It is actually a difficult question to address. We attempted to grow the strains in the presence of both daptomycin and actagardin but the results were difficult to interpret since both AMPs inhibit the growth of the strains. Therefore we did not claim that daptomycin inhibit AMP resistance.

4. Please provide more details on purification, reconstitution in proteoliposomes and ATPase assays since references lead to another reference.

We agree with the comment of the reviewer. The full experimental details of the reconstitution in proteoliposomes and ATPase assays are now provided in the revised manuscript. The purification protocol was well described in Diagne et al 2022, and we kept this reference.

Re: Spectrum03638-23R1 (Daptomycin avoids drug resistance mediated by the BceAB transporter in *Streptococcus pneumoniae*)

Dear Dr. Orelle,

Thank you for submitting your revised manuscript and for carefully responding to the reviewer's comments. I am pleased to accept your manuscript and I am forwarding it to the ASM production staff for publication. Your paper will first be checked to make sure all elements meet the technical requirements. ASM staff will contact you if anything needs to be revised before copyediting and production can begin. Otherwise, you will be notified when your proofs are ready to be viewed.

Sincerely,
Eric Cascales
Editor
Microbiology Spectrum